# Antifouling Polyethersulfone-Petrol Soot Nanoparticles Composite Ultrafiltration Membrane for Dye Removal in Wastewater

**DOI:** 10.3390/membranes11050361

**Published:** 2021-05-15

**Authors:** Nkechi P. Nwafor, Richard M. Moutloali, Keneiloe Sikhwivhilu, Oluwole B. Familoni, Luqman A. Adams

**Affiliations:** 1Materials and Nanochemistry, University of Lagos, Lagos 101212, Nigeria; nkynwafor_71@yahoo.com (N.P.N.); ladams@unilag.edu.ng (L.A.A.); 2DSI/Mintek Nanotechnology Innovation Centre—UJ Water Research Node, University of Johannesburg, Doornfontein, Johannesburg 2028, South Africa; 3DSI/Mintek Nanotechnology Innovation Centre, Advanced Materials Division, Mintek, 200 Malibongwe Drive, Randburg, Johannesburg 2125, South Africa; KeneiloeS@mintek.co.za; 4Organic Chemistry, University of Lagos, Lagos 101212, Nigeria; familonio@unilag.edu.ng

**Keywords:** oxidized petrol soot, membranes, polyethersulfone, fouling reduction, dye rejection

## Abstract

Engineered nanoparticles are known to boost membrane performance in membrane technology. Hitherto, tunable properties that lead to improved hydrophilicity due to increased surface oxygen functionalities upon oxidation of petrol soot have not been fully exploited in membrane filtration technology. Herein, the integration of oxidized petrol soot nanoparticles (PSN) into polyethersulfone ultrafiltration membranes produced via phase inversion technique for dye removal in wastewater is reported. The nanoparticles, as well as the composite membranes, were characterized with diverse physicochemical methods, particularly TEM, SEM, BET, AFM, contact angle, etc. The effect of varying the ratio of PSN (0.05–1.0 wt %) on the properties of the composite membrane was evaluated. The composite membranes displayed increased hydrophilicity, enhanced pure water flux, and antifouling properties relative to the pristine membrane. For example, the obtained pure water flux increased from 130 L·m^−2^·h^−1^ for base membrane to 265 L·m^−2^·h^−1^ for the best composite membrane (M4). The best flux recovery ratio (FRR) observed for the membranes containing PSN was ca. 80% in contrast to 49% obtained with the pristine membrane indicative of the positive influence of PSN on membrane antifouling behavior. Furthermore, the PSN composite membranes displayed relatively selective anionic dye rejection of ˃95% for Congo red and between 50–71% for methyl orange compared with 42–96% rejection obtained for cationic methylene blue dye with increasing PSN content. The successful fabrication of polyethersulfone–PSN composite membranes by a simple blending process opens a novel route for the preparation of economical, functional, and scalable water purification membranes capable of addressing the complex issue of water remediation of organic azo dyes.

## 1. Introduction

The depletion of clean water resources due to drought and population growth has necessitated more emphasis on wastewater reuse [1]. Extensive industrialization with concomitant lack of treatment of industrial effluents has increased the rate of pollution of water bodies with diverse pollutants such as suspended solids, dyes, pesticides, fertilizers, toxic metals, etc. [2,3]. Dyes, predominantly utilized in food, drug, cosmetic, and textile industries, are toxic, generally non-biodegradable and recalcitrant, affecting the chemical oxygen demand (COD) as well as decreasing the light infiltration in water bodies, thus having an adverse consequence on water organisms [1,2,4]. Consequently, several techniques have been deployed for dye removal including the use of coagulants, adsorption, photo-degradation, membrane filtration, ozonation, electrochemical-oxidation, and chemical/biological degradation [4]. The requirements for compliance to stringent regulations regarding wastewater disposal into water bodies, as well as the high cost of water, have compelled industrial facilities to revisit their residual water management strategies [5] with the focus on treatment methods with high removal efficiencies for dyes, ease of operation, low energy consumption, and high efficiency.

In this regard, ultrafiltration membranes have been demonstrated to be a viable approach as a result of their inherent benefits, which include relatively minimal energy and chemical consumption, as well as being environmentally friendly and offering a small footprint [6,7,8]. Nonetheless, the major drawback of this technology is the decline in permeate flow rates attributable to fouling [6,7]. Membrane fouling, which occurs either through cake formation or adsorption as well as deposition of foulants on the membrane, [9,10,11] negatively affects its performance indicators. These include reduction in membrane productivity leading to longer filtration times and reduced membrane life span due to material degradation, which is a consequence of harsh cleaning chemicals used that also alter the exclusion capacity of membranes due to changes in pore size [12].

Increasing the surface hydrophilicity of membranes is extensively accepted as an efficient means of enhancing the resistance of the membrane towards fouling [7,11,12]. This is because of the development of a tight water skin on the outer layer of the membrane resulting in decreased interaction with the foulants and membrane-selective layer [11]. Surface hydrophilicity is imparted through various approaches, these include skin grafting of monomers, integration of biomolecules, coating, and plasma treatment or exposure to ultraviolet radiation [7,11]. Nonetheless, these techniques are limited by excessive cost, low durability (which may destroy the surface structure), preferential modification of membrane surface, and inaccessibility [7,11]. Considerable reports have suggested that blending an organic polymer with fillers in dope solutions during the course of membrane fabrication can enhance the membrane hydrophilicity; hence, reducing the inherent membrane fouling through a simple procedure that requires no additional step [11,12]. Incorporation of additives into a dope solution additionally modulates the pore size and porosity of the membrane, improving the formation of macrovoids. Inclusion of nanoparticles to engineer the surface hydrophilicity of membranes is widespread [8,9,11,12] as a consequence of exploitation of their large surface-area-to-volume ratio in addition to chemical and mechanical resistance. Soot, obtained as waste or a byproduct of the incomplete combustion of fuels such as gasoline, kerosene, or diesel at a low temperature or limited supply of oxygen, is sized within the nanometer range [13], and can be a good candidate to be used as a nanofiller in polymer membranes. However, its hydrophobicity increases its tendency to aggregate with concomitant reduction in surface area and usability [14]. Nevertheless, its hydrophilicity can be improved by chemical modification through oxidation [14], e.g., using HNO_3_, an efficient oxidant whose oxidizing ability increases with increase in molarity and acidity, to oxidize the carbon walls, creating cracks for the insertion of carboxyl functional groups [14,15,16].

Fabrication of low fouling composite membranes by incorporating oxidized petrol soot nanoparticles into polyethersulfone matrix via phase inversion is reported here for the first time. The morphology and structure of composite membranes in addition to their performance assessment through pure water flux, flux recovery, and dye removal capacity were investigated, and the results are detailed herein. This demonstrates the potential of utilizing low-cost sources of carbon such as soot to enhance membrane performance whilst decreasing overall cost of membrane components.

## 2. Materials and Methods

### 2.1. Materials

All commercial reagents used were procured from Sigma Aldrich, South Africa, and utilized as supplied with the exception of polyethersulfone (PES) (Mw ≈ 58,000 g·mol^−1^), which was obtained from Protea Laboratories, South Africa. N-methylpyrolidone (NMP) and polyvinylpyrrolidone (PVP, Mw ≈ 40,000 g·mol^−1^) were employed as solvent and pore former, respectively. Methylene blue (MB, molecular weight ≈ 320 g·mol^−1^), methyl orange (MO, molecular weight ≈ 327 g·mol^−1^), and Congo red (CR, molecular weight ≈ 697 g·mol^−1^) were utilized for membrane rejection studies. Bovine serum albumin (BSA, MW = 66,000 Da) was used for fouling studies and HNO_3_ (70% wt) for oxidation of soot. Petrol (98 premium unleaded PMS) soot was obtained from motor vehicle exhaust in Nigeria, oxidized with HNO_3_, and subsequently used as inorganic membrane fillers.

### 2.2. Oxidation of Petrol Soot

Petrol soot was oxidized using established protocols developed for the oxidation carbon black [17,18,19]. In short, petrol soot (10.00 g) was dispersed in HNO_3_ (70%, 1:10 *w/v*) overnight, heated at 80 °C, then cooled to ambient temperature and subsequently centrifuged at 7000 rpm. The resultant pellet was rinsed in plentiful amounts of deionized water to a neutral pH followed by drying at 110 °C overnight to afford oxidized petrol soot nanoparticles (PSN).

### 2.3. Preparation of Membranes

Polymer composite membranes were fabricated using an established phase inversion technique induced by immersion precipitation [20,21,22]. Polyethersulfone (PES) was dried at 100 °C overnight prior to use. Firstly, a specific mass of PSN (with respect to the gross mass of the solution) was dispersed in *N*-methylpyrolidone (NMP) by sonicating for 30 min to afford a well-dispersed nanoparticle suspension. Distinct amounts of polyethersulfone (PES), the base polymer, and polyvinylpyrrolidone (PVP), the pore forming agent, were then added and allowed to dissolve under constant stirring at 600 rpm overnight at ambient temperature to attain good homogeneity of nanoparticles in the precursor solutions. Solution composition is contained in Table 1, where for each formulation three casting solutions were made producing three membranes. Two samples were obtained from each membrane and assessed under the same conditions. Entrapped air bubbles in the resulting homogeneous polymer solutions were eliminated by degassing in a vacuum overnight before the casting solution was spread on a clean glass plate fitted with an automated stainless-steel blade, maintaining an air-gap of 150 µm. It is noteworthy that the resultant suspension did not exhibit clumps and remained black when left overnight under reduced pressure without the development of solid deposits during the degassing stage. The cast polymeric film was submerged in a coagulation bath containing deionized water after 30 s exposure in air. It was allowed to stand in the coagulation bath for 15 min, during which the formed membrane detached from the glass plate. The membrane film that subsequently peeled off was preserved overnight in fresh deionized water to ensure total solvent and pore former elimination from the membrane film. Fabricated membranes were preserved in deionized water prior to utilization.

### 2.4. Physicochemical Characterization of PSN Nanoparticles and Membranes

Surface properties of PSN were ascertained using transmission electron microscopy (TEM, JOEL JEM–2100, Tokyo, Japan), whose field emission gun was operated at 200 kV. The nanoparticles were dispersed in about 5 mL of methanol for 10 min prior to being deposited on a carbon coated copper grid, and subsequently dried at room temperature and then mounted on the exchange rod of the TEM chamber prior to investigation.

Functional group analysis on the soot nanoparticles was investigated by Fourier-transform infrared spectroscopy (FTIR, Perkin Elmer Spectrum 100, Waltham, MA, USA), operated within 500–4000 cm^−1^ scan range and averaged over 16 scans to reduce spectral noise. Dried PSN was ground with potassium bromide (KBr) in 1:10 ratio and pressed into pellet form prior to FTIR investigation.

The thermal stability of the nanoparticles was examined with a thermogravimetric analyzer (TGA, Hitachi, STA7200 RV, Tokyo, Japan). The nanoparticles (0.991 mg) were heated from 25–700 °C at a heating rate of 10 °C min^−1^ in a flowing nitrogen environment (20 mL·min^−1^).

Using an automated gas adsorption analyzer, physisorption analysis was performed with an ASAP 2020 Micromeritics instrument (Dublin) utilizing the Brunauer–Emmett–Teller (BET) analysis to determine the surface area and pore volume of PSN. About 0.2 g of PSN was degassed for 4 h in the Micromeritics degassing unit at 150 °C under a flowing nitrogen atmosphere (60 cm^3^.min^−1^) for 4 h.

The morphological characteristics of membranes were examined by scanning electron microscopy (SEM, TESCAN VEGA 3, a.s., Brno, Czech Republic) operated at 20 kV voltage. To obtain the cross-sectional structure of the membranes, air-dried membrane samples were dipped in liquid nitrogen and broken prior to analysis. Membrane samples were carbon-coated before analysis.

The membrane’s surface hydrophilicity was determined through contact angle measurements using a drop-shaped contact angle analyzer (DataPhysics, SCA 20, KRUSS, Hamburg, Germany) equipped with a video camera. A microliter of deionized water was dropped on the membrane outer layer and the contact angles measured. Measurements were taken on three replicates on at least five to ten different sites, averaged to minimize experimental error.

Study of the membrane topography was accomplished with an atomic force microscope (AFM, Nanoscope IV, Veeco Metrology Group, California, CA, USA), operating in a non-contact mode. The topography and other surface properties were measured over a larger area at different locations to asses sample uniformity. Once completed, specific smaller sections were scanned at higher resolution.

### 2.5. Membrane Performance Tests

The filtering capability of membranes was ascertained at 20 ± 1 °C utilizing a dead-end stirred cell filtration system (Sterlitech HP 4750, Washington, USA), having an area and capacity of 12.6 cm^2^ and 250 mL, respectively, with applied pressure derived from compressed nitrogen supply line. Membranes were preliminarily dipped in deionized water for 30 min and compacted using deionized water at 150 KPa for 30 min to stabilize the permeate flux and reduce fluctuation during filtration tests [23]. The pure water flux and rejection performance of membranes were determined at 100 KPa. Collection and measuring of permeates was done every 5 min. The water flux was deduced from Equation (1) [7,21,24,25,26]:(1)$$J_{0\;}=\frac{V}{A\Delta t}\;$$
where *J_o_* (L·M^−2^h^−1^) = pure water flux, *V* (L) = volume of the pure water permeates, *A* (M^2^) = membrane effective area, and Δ*t* (h) = time of permeate collection. Triplicate measurements were obtained on three different membranes and the average taken for each membrane to minimize experimental errors. After pure water filtration, 1 g·L^−1^ of bovine serum albumin (BSA) solution was filtered through the membrane under the same conditions as the pure water flux filtration. BSA permeate flux was denoted as *J*_1_. Thereafter, the fouled membranes were rinsed in distilled water for 20 min to simulate backwashing process before being returned to the cell. The water flux of the cleaned membranes (*J*_2_) was obtained following similar protocols as detailed above. The membrane antifouling characteristics were calculated from the flux recovery ratio (*FRR*) as given by Equation (2) [21,23,24,25]:(2)$$FRR=\frac{J_{2\;}}{J_{o\;}}\;\times \;100\%$$

The flux loss as a result of reversible (*R_r_*) and irreversible (*R_ir_*) membrane protein fouling as well as the total membrane fouling (*R_t_*), which depicts the extent of flux decline resulting from the combined protein fouling in the membrane, were derived from expressions (3), (4) and (5), respectively [21,23].
(3)$$R_{r}=\left(\frac{J_{2}-J_{1}}{J_{0}}\right)\;\times 100\%\;$$
(4)$$R_{ir}=\left(\frac{J_{0}-J_{2}}{J_{0}}\right)\;\times \;100\%$$
(5)$$R_{t}=R_{r}+R_{ir}=(1-\frac{J_{1}}{J_{0}})\;\times \;100\%\;$$

Dye rejection capacity of membranes with respect to methyl orange, methylene blue, and Congo red dyes were also investigated. Fresh samples of membranes were cut and compacted as described above. Afterwards, the reservoir of the dead-end filtrating unit was emptied and refilled using an appropriate dye solution. Aqueous feed solutions (100 mL, 30 mg·L^−1^) of dyes in simulated textile wastewater was passed through the membranes using a working pressure of 100 KPa at ambient conditions. The pressure used in the current study was at the lower end of the applied pressure range of the UF filtration and was selected based on practical considerations, i.e., small volume dead-end cell used. Stirring was employed to minimize concentration polarization during measurement [26]. Dye permeate (10 mL) was collected from each membrane. Quantitative analysis of the dye concentration in both feed and permeate was determined using ultraviolet–visible spectrophotometry (Shimazdu, Japan). Dye rejection capacity of membrane R (%) is calculated using Equation (6) [4,20,26]:(6)$$R\left(\%\right)=\left(1-\frac{C_{p}}{C_{f}}\right)\;\times \;100$$
in which *C_p_* and *C_f_* represents dye concentrations in permeate and feed.

## 3. Results

### 3.1. Characterization of Oxidized Petrol Soot Nanoparticles

TEM images of the PSN before (Figure 1A) and after oxidation (Figure 1B) showed that the nanoparticles were 13–22 nm before oxidation with a high degree of agglomeration, which decreased to a narrow size distribution (2–8 nm) with reduced aggregation after oxidation. Aggregation of HNO_3_ oxidized carbon black and soot particles was previously described by Marcelo et al., [19] and Rosca et al., [18] respectively. Hydrophobicity of soot particles increases their tendency to aggregate [14,27,28] with concomitant reduction in surface area and usability as nanofillers. However, chemical oxidation of hydrophobic soot particles minimized their aggregation and improved their dispersibility [14]. Oxidation led to increased surface functional groups that increased interaction with polar solvents, methanol in this case, which led to reduced particulate aggregation observed in the present case.

The successful oxidation of the soot particles was confirmed by the increased amount of oxygenated functional groups (including carboxylic and hydroxyl) observed from the FTIR spectra (Figure 2) [14,19]. Specifically, the appearance of bands at 1722 cm^−1^, 1545 cm^−1^, and 1384 cm^−1^ representing C=O, C–C, and C–O stretch vibrations, respectively, are indicative of carboxylic group. The band observed at 1384 cm^−1^ signifies the emergence of oxygenated moieties either from hydroxyl, carboxyl, or carbonyl as a result of the oxygenation [14], whilst the broad band at 3433 cm^−1^ is as a result of O–H stretch, further confirming the presence of carboxylic acid groups.

Thermogravimetric analysis (Figure 3a) exhibited three thermal events. The initial event between 50–150 °C is assigned to the dehydration of the samples [15]. The loss from 200–350 °C is assigned to the degradation of the carboxylic group, whilst the loss from 350–580 °C is assigned to the loss of the hydroxyl functional group [29]. These support the successful oxidation of soot since these were absent in the nascent soot particles.

Surface area, pore diameter, as well as the pore volume of oxidized PSN as obtained from BET analysis were 2.4097 m^2^·g^−1^, 24.3869 nm, and 0.0203 cm^3^·g^−1^, respectively. The pore diameter of PSN is above 2 nm indicating that the material is mesoporous [30]. The observed surface area and pore volume obtained is slightly lower than the value, 5.396 and 1.239 respectively, reported by Joseph et al., [13] for nascent soot particles. This shortfall was ascribed to the resultant effect of introducing oxygenated groups into the carbon pore structure during HNO_3_ oxidation of carbon [19]. Figure 3b shows the nitrogen sorption isotherm for PSN which exhibited a type III isotherm, curving outward to the relative pressure axis, signifying that the quantity adsorbed increases with an increase in relative pressure [30]. These observations are in line with prior reports [13].

### 3.2. Membrane Characterization

#### 3.2.1. Characterization of Membranes, SEM

Figure 4 shows that the membrane surfaces were dominated by porous structures as expected of ultrafiltration membranes produced by phase inversion method. The micrographs revealed that there was a remarkable rise in abundance of pores on the composite membrane outer layer with respect to the neat membrane on PSN loading. It is noteworthy that the pore size comparison of the membranes was not interrogated at this stage as the Congo red probe molecule will be used to obtain relative rejection profiles. The increase in pore density is generally attributed to the inclusion of hydrophilic PSN moiety to the dope solution. Additionally, as the amount of soot increased, bright spots indicative of the soot nanoparticles on the surface increased, with pronounced agglomeration observed for the highest soot loading (M_5_). This could be attributable to the increased migration of the hydrophilic PSN particles at higher loading from the PES matrix closer to the surface to minimize the interfacial energy within the dope solution and the water surface in the course of phase inversion [23]. In addition, increased viscosity of a casting solution on greater filler amount decreases the process of phase inversion/separation, thus promoting clustering of nanoparticles as a result of reduced demixing rate [22,31]. The effect of this on membrane performance indicators is discussed in subsequent sections.

Micrographs of the cross-section (Figure 5) of fabricated membranes reveal a classic asymmetric arrangement consisting of a skin layer and a thick finger-like spongy sub-layer in all membranes. The addition of oxidized PSN to the dope solution remarkably resulted in an increased abundance of a finger-like microvoid sublayer in M_1_ (0.05 wt %) with respect to the base membrane (M_0_), signifying an increase in membrane porosity. On further increasing PSN loading in the dope solution (0.1–0.5 wt %), the finger-like microvoid sublayer enlarged across the membrane, signifying a further increase in porosity of the membrane relative to the base membrane. This is ascribed to the rise in the extent of substitution of solvent with water during the phase inversion process, a consequence of increased hydrophilicity of the dope solution by nanoparticles [24,32]. Nonetheless, excessive loading of PSN in M_5_ (1.0 wt %) decreased the abundance of the finger-like microvoid as heightened viscosity of dope solution started to dominate the demixing operation, suppressing the formation of the finger-like microvoid [24].

#### 3.2.2. Membrane Surface Roughness Characteristics

AFM was deployed in determining the roughness parameters of the fabricated membranes. The micrographs (Figure 6) reveal that the membrane surface became relatively smoother with the inclusion of PSN in the formulation. All composite membranes had lower roughness than the pristine membrane, with the lowest roughness observed for M_1_ (0.05 wt %), drastically reduced from 47.0 to 24.0 relative to the pristine membrane (M_0_). This decrease (M_0_ to M_1_) is ascribed to the good dissipation of PSN along with their migration towards the membrane surface in the course of phase inversion operation, leading to the formation of a more uniform and tighter surface with decreased surface roughness [22,33]. However, on further increasing the PSN concentration in the formulation, the surface roughness increased as a result of concomitant surface accumulation and aggregation of PSN particles [22,33]. Table 2 summarizes the roughness factors (*R_a_* and *R_q_*) obtained from analysis of AFM data. The data confirms that all composite membranes had lower roughness than pristine membranes which bodes well for potential fouling resistance of the composite membranes. Surface roughness is also significant in the fouling propensity of membranes [33]. The prevailing theory is that increasing surface roughness increases the contact area accessible for foulant interactions, resulting in considerable adhesion forces between the foulants and membrane [22,33,34,35].

#### 3.2.3. Membrane Contact Angle Measurement

Surface hydrophilicity was determined using water contact angle (CA) measurements (Figure 7a). The base membrane had a contact angle of 70°, indicative of its relatively hydrophobic character [24]. The contact angle decreased in composite membranes with increased PSN content, from a high of 65° to a low of 51° (M_4_, 0.5 wt %). It is widely reported that incorporation of hydrophilic fillers in polymer matrices improves membrane hydrophilicity [12,22,24]. It is noteworthy however that increasing PSN content to 1.0 wt % increased the contact angle to 57°. This increase in CA can be attributed to surface inhomogeneity and PSN agglomeration affecting membrane structure and character at this higher loading.

#### 3.2.4. Membrane Water Permeability

The effect of PSN concentration in the membrane on water flux is presented in Figure 7b. Pure water flux increased with increasing PSN concentration. Pure water flux rose from 133.06 L·m^−2^h^−1^ in the pristine membrane to 263.75 L·m^−2^h^−1^ in M_4_ (0.50 wt %). The sequence was increasing to 180.59 L·m^−2^h^−1^, 209.11 L·m^−2^h^−1^, 234.45 L·m^−2^h^−1^, and 263.75 L·m^−2^h^−1^ for M_1_ (0.05 wt %); M_2_ (0.10 wt %); M_3_ (0.30 wt %); and M_4_ (0.50 wt %), respectively. However, there was a slight decline in flux when PSN concentration rose from 0.5 wt % to 1.0 wt %, with a flux of 222.53 L·m^−2^h^−1^ recorded for M_5_ (1.00 wt % PSN). The observed increase in flux with increasing PSN is ascribed to a rise in membrane permeability, increased pore density, and hydrophilicity with increased nanoparticle dosage [22]. Additionally, the enlarged finger-like pore structure and their interconnectivity aided the water flux by grossly decreasing the membrane hydrophilic resistance. The direct relationship between pure water flux and hydrophilicity (as measured through contact angle) indicated that membrane hydrophilicity character plays a dominant role in membrane pure water flux [36]. The observed flux decline at the highest PSN loading is attributed to a decrease in membrane hydrophilicity and average pore density as a result of pore filling by the nanoparticles [22] and suppression of pore formation at high casting solution viscosity.

#### 3.2.5. Membrane Antifouling Characteristics

Severe fouling in PES-based membranes from hydrophobic reactions of membrane surface interactions with foulants and the concomitant membrane flux decline has been widely reported [12,24,25,37]. Membrane fouling leads to degeneration of the selectivity and permeability of the membrane during protein separation and reduces the membrane lifetime. Herein, bovine serum albumin was used as an artificial foulant to determine the antifouling capabilities of pristine and composite membranes. Figure 8 presents the evolution of the flux with filtration time for the as-fabricated membranes. In stage I, all the membranes showed a similar trend in pure water flux, showing an effective membrane compaction process [11]. However, there was a sharp decrease in flux when pure water was replaced with 1 g·L^−1^ bovine serum albumin solution in stage II in all the membranes. This is due to the deposition of the protein molecules and subsequent obstruction of the pores [11,24]. The BSA permeate dropped dramatically in the first 35 min of filtration before equilibrating. This is because the protein molecules from BSA feed deposited and adsorbed onto the membrane outer layer (cake formation and its participation in the process) during the ultrafiltration process, leading to a sharp flux decline at the early phase of the operation [23]. In stage III, the composite membranes displayed higher fluxes than the pristine membrane after mere hydraulic cleaning. Figure 8a depicts that the pristine membrane (M_0_) possessed the least flux recovery ratio (FRR) of 48.61%, while the composite membranes incorporated with PSN nanoparticles gave a higher FRR value. The highest FRR of 79.73% was observed in the M_4_ membrane incorporated with 0.5 wt % nanoparticles. However, increasing the loading of PSN up to 1.0 wt % decreased the FRR value to 59.48%, which was remarkably still higher than that obtained for the pristine membrane. From the FRR values, it is apparent that incorporation of PSN improved the hydrophilicity of the composite membranes, making accumulation as well as absorption of protein molecules on the outer layer of the membrane more difficult [24].

For a more detailed analysis of the antifouling characteristics of membranes, the reversible (*R_r_*), irreversible (*R_ir_*), as well as the total fouling ratios (*R_t_*) of all membranes were evaluated using Equations (3)–(5), with BSA containing aqueous feed solution. For reversible fouling, foulants are substantially effaced by hydrolytic cleaning, while irreversible foulants are effaced through chemical disinfection processes. Total fouling resistance is the sum of *R_ir_* and *R_r_*. Figure 9 shows that incorporating PSN into the PES membrane lowered the R_ir_ values from 51.39% in the pristine membrane to 20.27% in the M_4_ (0.50 wt %) membrane. In practice, this implies a reduction in the number of cleaning cycles (lower cleaning cost) required for composite membranes having a lower *R_ir_* [37]. The *R_r_* values of the membranes incorporated with PSN were greater than those of the pristine membrane, signifying an enhanced effacing of foulants from the outer layer of the composite membrane through a mere hydraulic disinfection operation [38]. The M_4_ membrane gave the highest *R_r_* value of 36.8%, while the pristine membrane gave the least *R_r_* value of 20.10%. The increased hydrophilic properties of the composite membranes (Figure 7a) resulted in reduced deposition of BSA and its irreversible interaction with the membrane surface and pores. Table 3 summarizes the fouling characteristics (FRR, *R_t_*_,_
*R_ir_* and *R_r_* values) of membranes.

#### 3.2.6. Membrane Dye Rejection

The dye exclusion capacity of fabricated membrane was determined using three model organic dye compounds, namely methyl orange (MO, MW ≈ 327 g·mol^−1^), methylene blue (MB, MW ≈ 320 g·mol^−1^), and Congo red (CR, ≈697 g·mol^−1^) at ambient temperature. The three model organic dyes were selected as target pollutants of industrial and environmental relevance, being potentially toxic, chemically resistant, and non-biodegradable [2,38,39]. All the three compounds are heterocyclic aromatics with MB being cationic, MO anionic, and CR a neutral dye. CR is the largest molecule whilst MO and MB have similar molecular weights. Figure 10 and Table 4 show the rejection profile and data obtained from all the membranes, respectively. Congo red was rejected by all membranes (˃95% rejection). MB was rejected highly by membranes with high PSN content (all above 80% with M_3_ and M_4_ at 90% and 96%, respectively), with M_1_ (42%) and M_0_ (48%) having similar rejection profile for this dye. MO was the least rejected dye at between 50 and 71% by all membranes, with M_3_ and M_4_ showing the highest rejections at 68% and 71%, respectively. It is worth nothing that M_5_ showed a decline in rejection relative to M_4_ in all cases, i.e., for CR it was 100% to 99%, for MB, 96% to 88% and MO, 71% to 64%. The high rejection of CR is a confirmation that the fabricated membranes are UF, as CR is being continually utilized in evaluating UF membranes’ molecular weight cut off [30]. On the other hand, the increasing rejection observed for MB with increasing PSN content indicates that there is specific interaction between the dye and the PSN fillers, probably due to some electrostatic interaction between oxygenated functional groups on PSN and the positive charges on the dye. Shenvi et al., [4] reported that electrostatic forces of attraction of a negatively charged membrane outer layer allying with positively charged MB dye molecule was responsible for the increased dye exclusion capacity exhibited. It is thought that similar behavior was operating here, resulting in high MB exclusion on high PSN loading. The narrow rejection band among all the membranes for MO indicates that the rejection mechanism for all of them is predominantly similar, probably size exclusion. This is supported by the overlap of the rejection band for MO, where rejection was lowest and dominated by size exclusion more than the electrostatic interaction alluded above.

## 4. Conclusions

A series of novel polyethersulfone oxidized petrol soot nanoparticle (PSN) composite membranes with different amounts of PSN (0, 0.05, 0.1, 0.3, 0.5, and 1.0 wt %) were successfully prepared and reported for water treatment. The incorporation of oxidized PSN improved the membrane surface hydrophilicity which positively imparted on both pure water flux and BSA fouling resistance. Rejection of model dye compounds revealed that all membranes were ultrafiltration (CR rejection > 95%) and that the prevailing dye rejection mechanism for the largest dye and the anionic dye was size exclusion. In addition to size exclusion, electrostatic interaction between the embedded PSN and the cationic dye led to a further increase in its rejection and hence a secondary rejection mechanism was established. The successful fabrication of polyethersulfone–PSN nanocomposite membranes by a simple blending process opens a novel route for the generation of adsorbent membranes that can address the complex issue of water remediation of organic dyes using cheap fillers from waste sources.

## Figures and Tables

**Figure 1 membranes-11-00361-f001:**
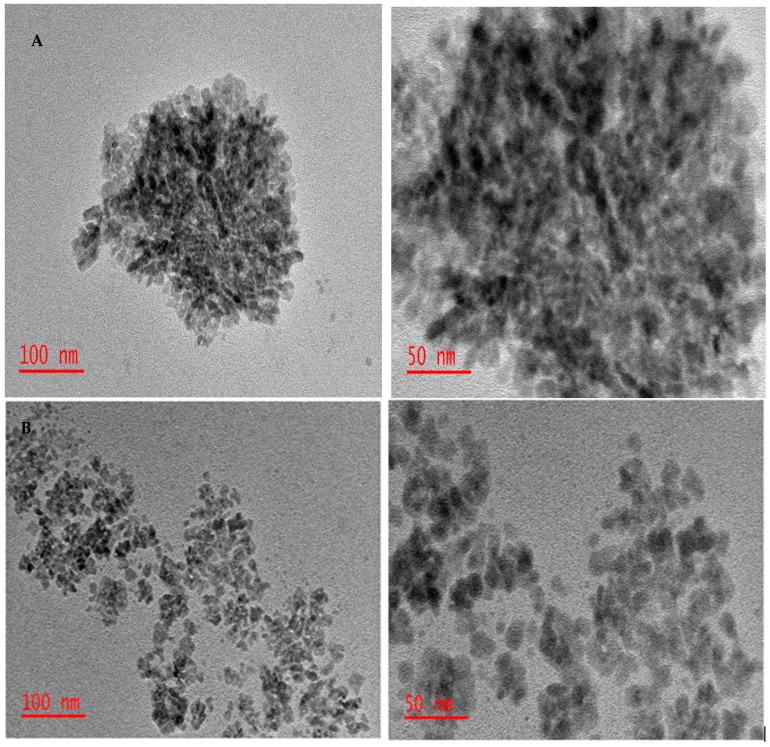
TEM images of the petrol soot nanoparticles before (**A**) and after oxidation (**B**).

**Figure 2 membranes-11-00361-f002:**
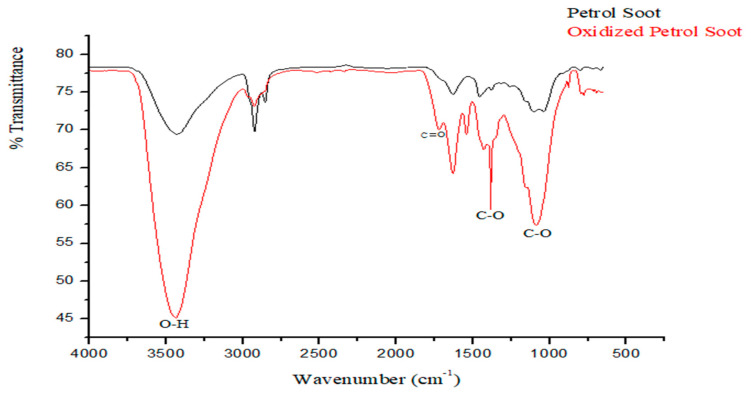
FTIR chromatogram of oxidized petrol soot nanoparticles.

**Figure 3 membranes-11-00361-f003:**
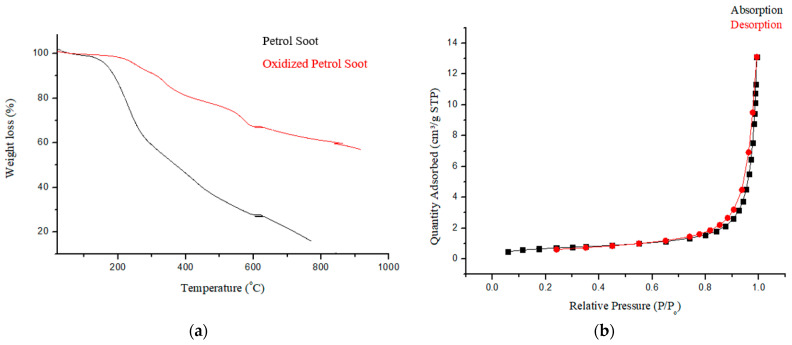
(**a**) Thermogram and (**b**) N_2_ adsorption isotherm of PSN showing type III isotherm.

**Figure 4 membranes-11-00361-f004:**
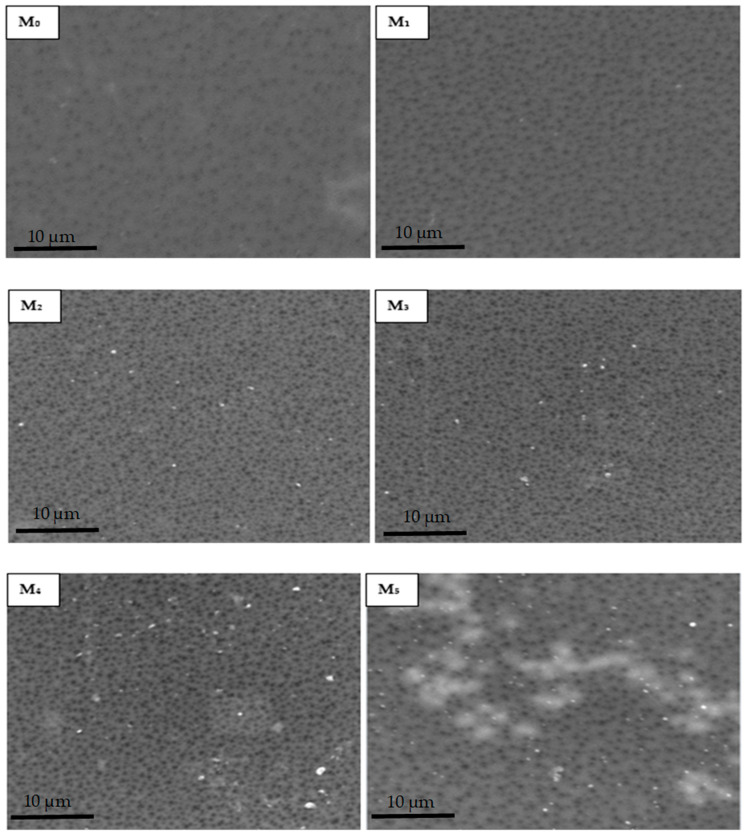
SEM micrographs of membranes with varying amounts of PSN: (M_0_) 0.00 wt %, (M_1_) 0.05 wt %, (M_2_) 0.10 wt %, (M_3_) 0.30 wt %, (M_4_) 0.50 wt %, and (M_5_) 1.00 wt %.

**Figure 5 membranes-11-00361-f005:**
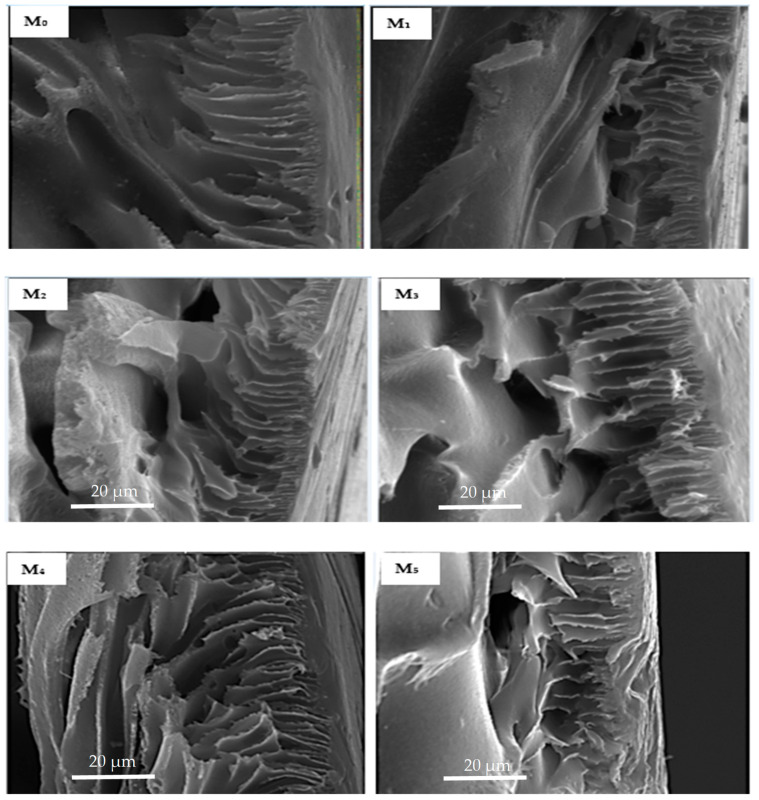
Cross-section SEM micrographs of membranes with varying amounts of PSN: (M_0_) 0.00 wt %, (M_1_) 0.05 wt %, (M_2_) 0.10 wt %, (M_3_) 0.30 wt %, (M_4_) 0.50 wt %, and (M_5_) 1.00 wt %.

**Figure 6 membranes-11-00361-f006:**
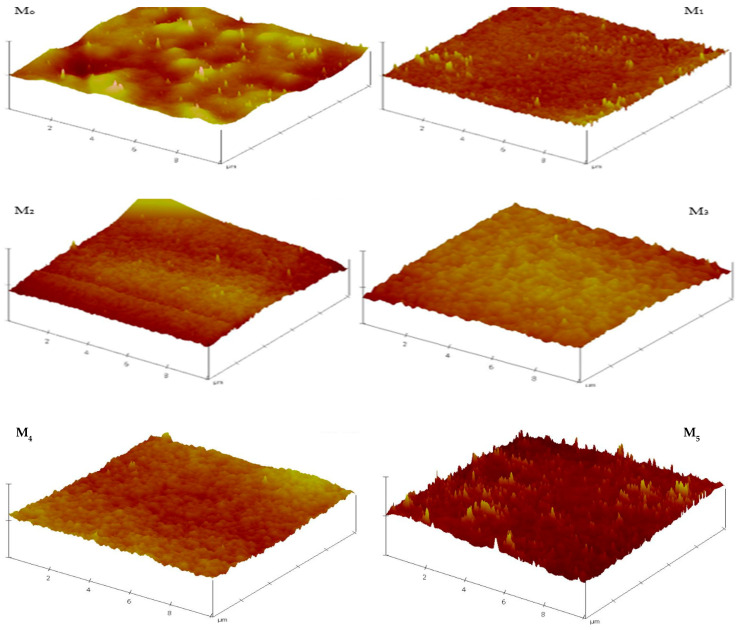
3D AFM micrographs of composite membranes depicting evolution of surface roughness with PSN content.

**Figure 7 membranes-11-00361-f007:**
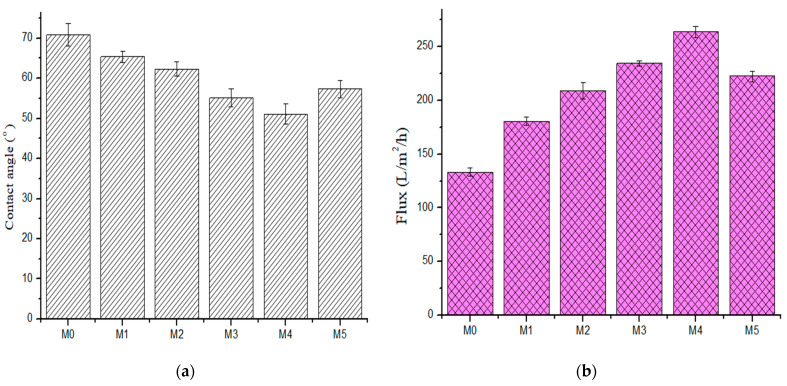
(**a**) Contact angle (**b**) pure water flux of fabricated membranes (M_0_) 0.0 wt %, (M_1_) 0.05 wt %, (M_2_) 0.1 wt %, (M_3_) 0.3 wt %, (M_4_) 0.5 wt %, and (M_5_) 1.0 wt %.

**Figure 8 membranes-11-00361-f008:**
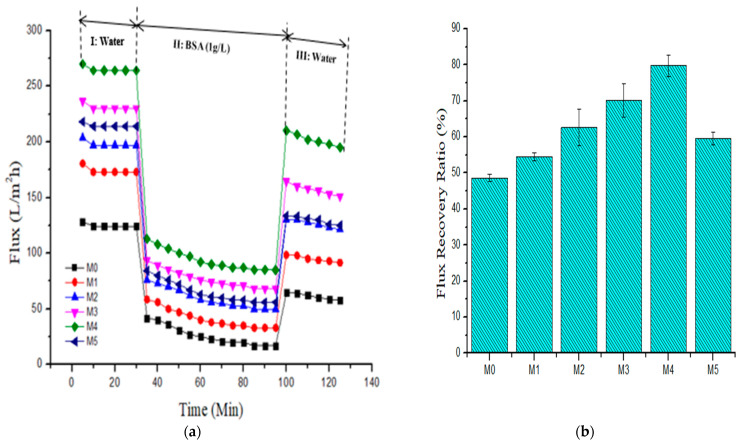
(**a**) Pure water flux and BSA (1 g·L^−1^) fouling pure water washing cycle and (**b**) resultant flux recovery ratio for fabricated membranes (M_0_) 0.0 wt %, (M_1_) 0.05 wt %, (M_2_) 0.1 wt %, (M_3_) 0.3 wt %, (M_4_) 0.5 wt %, and (M_5_) 1.0 wt %.

**Figure 9 membranes-11-00361-f009:**
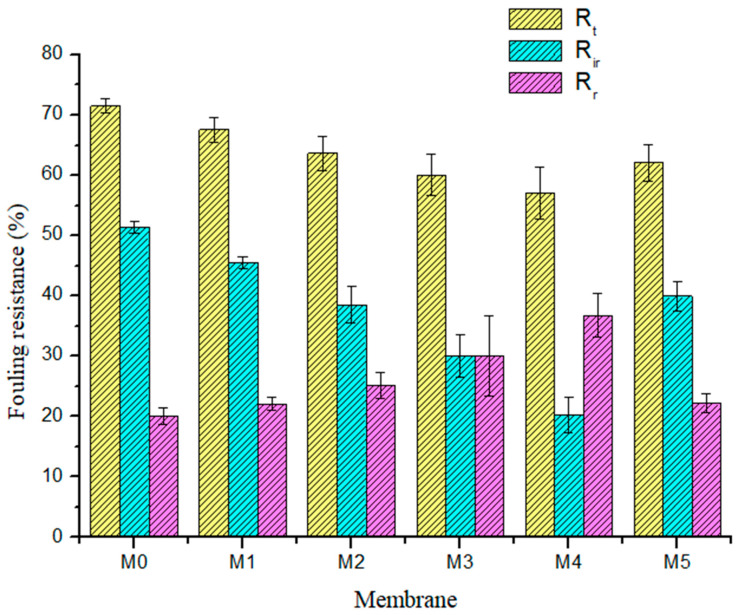
Summary of the membrane fouling types: total fouling ratio (yellow: *R_t_*), irreversible fouling ratio (blue: *R_ir_*), and reversible fouling ratio (purple: *R_r_*) for fabricated membranes.

**Figure 10 membranes-11-00361-f010:**
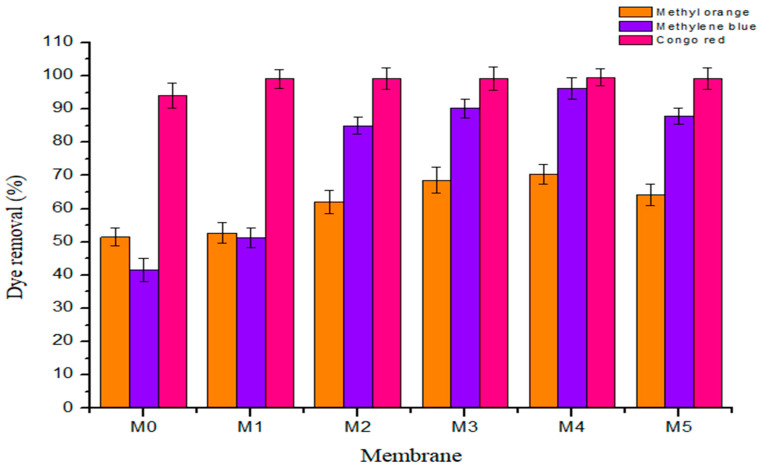
Rejection profile of the three heterocyclic aromatic dyes on fabricated membranes.

**Table 1 membranes-11-00361-t001:** Casting solution compositions.

Membrane ID	PES wt %	PVP wt %	PSN wt %	NMP wt %
M_0_	18.00	2.00	0.00	80.00
M_1_	18.00	2.00	0.05	79.95
M_2_	18.00	2.00	0.10	79.90
M_3_	18.00	2.00	0.30	79.70
M_4_	18.00	2.00	0.50	79.50
M_5_	18.00	2.00	1.00	79.00

**Table 2 membranes-11-00361-t002:** Surface roughness data of pristine and composite membranes.

Membrane ID	*R_a_*	*R_q_*
M_0_	47.0	60.5
M_1_	24.0	34.0
M_2_	43.4	55.4
M_3_	41.4	52.6
M_4_	35.7	44.8
M_5_	42.7	56.6

**Table 3 membranes-11-00361-t003:** Flux recovery ratio and fouling resistance data of pristine and composite membranes.

Membrane ID	FRR (%)	R*_t_* (%)	R*_r_* (%)	R*_ir_* (%)
M_0_	48.6	71.5	20.1	51.4
M_1_	54.5	67.6	22.1	45.5
M_2_	62.6	63.7	25.1	38.5
M_3_	70.2	60.1	30.1	30.0
M_4_	79.7	57.1	36.8	20.3
M_5_	59.5	62.1	22.1	39.9

**Table 4 membranes-11-00361-t004:** Rejection data of the three dyes on the fabricated membranes.

Membrane ID	Methyl Orange (%)	Methylene Blue (%)	Congo Red (%)
M_0_	51.5	41.5	94.0
M_1_	52.7	51.2	99.1
M_2_	62.1	85.0	99.2
M_3_	68.6	90.3	99.3
M_4_	70.4	96.3	99.5
M_5_	64.3	87.9	99.2

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
