# Peer review of "Antifouling Polyethersulfone-Petrol Soot Nanoparticles Composite Ultrafiltration Membrane for Dye Removal in Wastewater"

_membranes, 2021, doi:10.3390/membranes11050361_

Round 1

Reviewer 1 Report

The manuscript “Antifouling Polyethersulfone-Petrol Soot Nanoparticles Composite Ultrafiltration Membrane for Dye Removal in Wastewater”, by Nkechi P. Nwafor, Richard M. Moutloali, Keneiloe Sikhwivhilu, Oluwole B. Familoni, Luqman A. Adams, reports about the characterization and performance assessment of new composite membranes of polymers (polyethylenesulfone) and nanoparticles (oxidized petrol soot, PSN) for filtration technologies.

The membranes with different content of PSNs were characterized by a multitude of methods, such as TEM, SEM, BET, AFM, contact angle. The membrane properties and performance were assessed through pure water flux, flux recovery and dye removal capacity investigations, as a function of PSN content vs. unloaded membrane.

The manuscript is well written, the study is well designed and carried out in detail, the results are well presented and discussed.

I would have only one observation to make: Figure 3a – shouldn’t the TGA curves start at 100%? Why around 120%?

After clarification of this aspect, I would recommend the publication of the manuscript in Membranes.

Reviewer 2 Report

The authors fabricated PES composite membranes with different fractions of PSN and investigated their properties and filtration performance. The manuscript is in general well organized and written; the authors provided proper references to introduce the background of the work and gave detailed descriptions for experimental methodology. Some questions/comments regarding the results and discussion:

  • How many replicate samples were made for each PSN wt%? Were the results/trends reproducible?
  • Any characterizations done for the dispersity of PSN in NMP and the polymer solution? The authors mentioned that for TEM images shown in Figure 1, particles were dispersed in methanol before drying on the grid, and the dispersity can change significantly in different solvents.
  • There is no pore size characterization of the composite membranes. How big are the pores in the retentive layer and how does the incorporation of PSN affect the resulting pore size?
  • Regarding Figure 4, how does the aggregation of PSN affect the morphology and performance of membranes? Did the authors observed heterogeneity in terms of membrane performance?
  • On page 7 – 8, the authors stated that the porosity of membrane first increased then decreased with the amount of PSN added, any quantitative characterization of porosity done to support this statement?
  • Surface roughness can vary significantly from spot to spot – how many locations were scanned for each sample? If the aggregation of PSN can increase surface roughness, why M4 has lower surface roughness than M2 and M3?
  • An increase in surface roughness can make hydrophilic surface more hydrophilic (https://web.mit.edu/nnf/education/wettability/rough%20ideas%20on%20wetting.pdf). Therefore, was the decrease in contact angle caused by the increase of surface roughness? In line 266, why surface heterogeneity caused an increase in contact angle?
  • Do the authors think surface roughness is a significant factor that can affect membrane fouling (line 251 – 253)? If so, why M1 showed the worst anti-fouling property although it was the least rough?
  • How does the zeta potential of composite membrane change with the addition of PSN? This is important to understand the electrostatic interactions between charged dyes and the membranes.
  • Figures look sketchy. Most of them are obviously stretched in either the x direction or the y direction and look really unprofessional. Resolution of SEM images are low, texts are hardly readable.

Reviewer 3 Report

An economic nanomaterial soot was introduced in the ultrafiltration membrane composition that could potentially improve the membrane properties against fouling and dye removal. Membrane characterizations were well adopted, together with membrane performance testing.

  1. Line 58-76: What’s the cost of soot modification, compared with other membrane modification methods?
  2. The study aims at developing an anti-fouling membrane for wastewater treatment. Besides BSA as a model organic foulant, how important is bio-fouling and colloidal fouling?
  3. Section 2.5&3.2.5: Please describe the fouling experiment steps, with either an actual photo or a schematic drawing. Core question here is: the foulant/dye concentration is not constant through the testing and may affect the results.
  4. Typical UF operation pressure is 0.2-1.0 MPa. Why was the working pressure determined at 100 KPa in this study? What impact would it have compared with the higher pressure conditions?
  5. Line 172-177: What’s the size distribution of soot before oxidization? What improvement was made by the process and by how much? Please show the aggregation status before oxidation, and describe the criteria of aggregation level.
  6. Figure 4: To my understanding, the black spots represent the pores on the membrane surface. Thus, M5 has a much lower pore density compared with other PSN dosages. Please explain.
  7. What’s the range of pore size of the composited membranes? Does it fall in the UF range? What’s the molecular size of your model foulant and dyes? Is UF able to reject them? Is the rejection mechanism possibly dominated by charge exclusion? How to prove it?
  8. Section 3.2.2: It seems the surface roughness does not distribute evenly across the sample surface. The surface roughness of different membranes would make more sense with replicate tests and standard deviations. Line 247-249: M4 shows lower Ra and Rq. Please explain this reversed trend than M1-M3.
  9. What was the BSA rejection percentage for different membranes? Does the rejection rate change over time? The flux curve for all membranes flats out at ~85min operational time. What does it represent?
  10. Line 297-299: In dead-end flow mode, everything ends up on membrane surface. In this case, how much impact could membrane surface hydrophilicity make? In real NF application, cross-flow mode could make more use of the more hydrophilic surface. Thus, does the dead-end mode adopted in this study well represent the real case? If so, please explain. If not, what improvements could be made?

Round 2

Reviewer 2 Report

The authors addressed the reviewer's comments in the response letter; relevant discussion should also be added in the manuscript, including and not limited to particle dispersity and effect of surface roughness and hydrophilicity on fouling.
